# Echocardiographic view and feature selection for the estimation of the response to CRT

Alban Gallard[1], Elena Galli[1], Arnaud Hubert[1], Auriane Bidaut[1], Virginie Le Rolle[1], Otto Smiseth[2], Jens-Uwe Voigt[3], Erwan Donal[1‡]*, Alfredo I. Hernández[1‡]

1 University of Rennes, CHU Rennes, Inserm, LTSI UMR 1099, Rennes, France, 2 Center for Cardiological Innovation and Department of Cardiology, Oslo University Hospital, Oslo, Norway, 3 Department of Cardiovascular Sciences, KU Leuven, Leuven, Belgium

☯ These authors contributed equally to this work.
‡ These authors also contributed equally to this work.
* erwan.donal@chu-rennes.fr

**Data Availability Statement:** All relevant data are within the paper and its Supporting information files.

**Funding:** The author(s) received no specific funding for this work.

## Abstract

Cardiac resynchronization therapy (CRT) is an implant-based therapy applied to patients with a specific heart failure (HF) profile. The identification of patients that may benefit from CRT is a challenging task and the application of current guidelines still induce a non-responder rate of about 30%. Several studies have shown that the assessment of left ventricular (LV) mechanics by speckle tracking echocardiography can provide useful information for CRT patient selection. A comprehensive evaluation of LV mechanics is normally performed using three different echocardioraphic views: 4, 3 or 2-chamber views. The aim of this study is to estimate the relative importance of strain-based features extracted from these three views, for the estimation of CRT response. Several features were extracted from the longitudinal strain curves of 130 patients and different methods of feature selection (out-of-bag random forest, wrapping and filtering) have been applied. Results show that more than 50% of the 20 most important features are calculated from the 4-chamber view. Although features from the 2- and 3-chamber views are less represented in the most important features, some of the former have been identified to provide complementary information. A thorough analysis and interpretation of the most informative features is also provided, as a first step towards the construction of a machine-learning chain for an improved selection of CRT candidates.

## Introduction

Cardiac resynchronization therapy (CRT) is one of the most effective therapies for patients suffering from heart failure with reduced ejection fraction. CRT leads to improved quality of life and significant reductions in heart failure hospitalization rates and all-cause mortality [1, 2]. 20 to 30% of heart failure patients with reduced ejection fraction show a left bundle branch block (LBBB). These patients might benefit from a CRT implantation, with a reverse remodeling of the left ventricle (LV), an improvement in systolic function, a better control of symptoms and, finally, an improved quality of life and life expectancy [3]. Despite its

**Competing interests:** The authors have declared that no competing interests exist.

well-established clinical benefits and cost-effectiveness, it remains a widely underutilized treatment option. The topic of "non-response" to CRT has received large research attention. However, recommendations remain unchanged during the past years, mainly based on cardiac electrical activity markers, without the integration of novel markers related to the assessment of cardiac mechanical dyssynchrony and electro-mechanical couplingss [4, 5].

To overcome this important issue, a variety of methods have been proposed. One solution is to improve CRT implantation [6]. The optimal implantation of the LV lead is mainly based on the possibility to get an appropriate vein and good stimulation parameters. Therefore, most of the gain we could expect from CRT is expected to be in the selection process. Tremendous efforts have been made and a best understanding of mechanical dyssynchrony and electro-mechanical coupling have been achieved this recent years using imaging tools that are much more robust than in the past [7, 8]. Another solution aims at improving the identification and characterization of patients that are probably good responders to CRT. Among many methods, some authors have suggested that the analysis of speckle tracking echocardiography could improve the selection of CRT candidates [9–11]. The aim is to better characterize the regional mechanical function of the LV through this technique. In particular, some markers derived from LV wall motion estimation (peaks and timings) have been associated with CRT response. The analysis of LV deformation (strain) has also shown to be useful to understand LV mechanics and to provide information for CRT response estimation [5, 12, 13].

The best practice to assess overall LV mechanical deformation by bidimensional echocardiography is to acquire 3 different views (Fig 1) [14]: i) the 4 chambers view (4ch) which shows the two atria and the two ventricles; ii) the 2 chambers view (2ch) only focused on the left

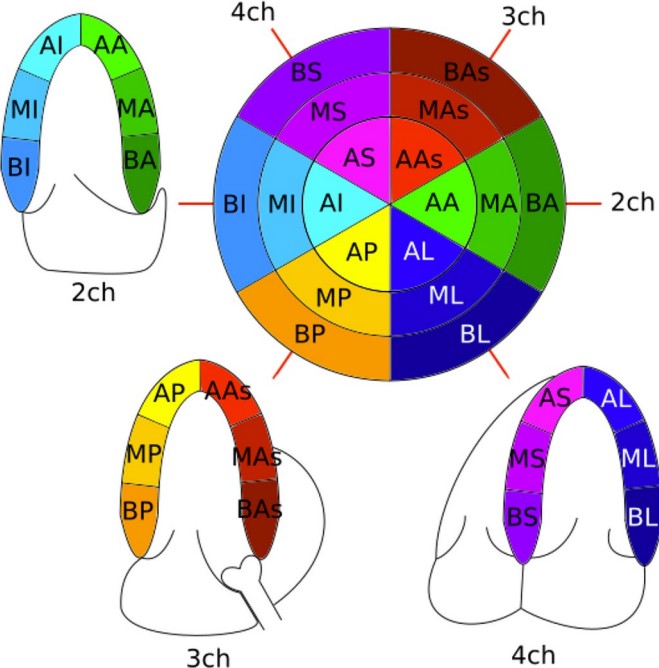

**Fig 1. Bull's-eye representation of myocardial LV segments and their relation to the different echocardiographic views (4ch: 4-chamber, 2ch: 2-chamber, 3ch: apical long axis).** In each label, the first letter corresponds to B: Basal, M: Mid, and A: Apical. The second letter correspond to S: Septal and L: Lateral for the 4ch, I: Inferior and A: Anterior for the 2ch, P: Posterior and As: Anteroseptal for the 3ch.

atrium and the left ventricle and iii) the apical long axis view (3ch) focused on the left part of the heart with the aorta.

Data acquired using these views can be processed to extract a wide variety of features. Machine Learning (ML) methods can then be applied to process these features in order to estimate the response to CRT, as shown in a recent work from our group, that was initially limited to the 4ch view [15]. Other works have also applied ML to predict the response to CRT or outcomes after CRT implantation [16, 17]. But a critical step in this approach is an appropriate feature analysis study, which implies the estimation of the relative importance of each input feature to the global decision output. Moreover, this feature analysis may lead to a feature selection step in which the number of features used is reduced while maintaining an acceptable performance level. By reducing the dimension of the problem in such a way, the convergence and generalization properties of the ML method are improved, while minimizing overfitting [18, 19].

Concerning the estimation of CRT response, a recent work by Cikes et al. applied an unsupervised method based on Multiple Kernel Learning and K-means clustering to a set of 1106 HF patients [13]. After the clustering step, the authors identified four natural groups and characterized these groups through expert knowledge according to their phenotypes. This data mining approach allowed for the identification of two phenogroups presenting a higher proportion of some clinical characteristics that are known from the literature to be predictive of CRT response. In this work, we propose another view to the problem, directly focused on the quantitative analysis of myocardial deformation for the selection of the most informative echocardiographic views and features for the estimation of CRT response. To this end, we apply a supervised approach, based on the Random Forest (RF) method and focused on a set of advanced, quantitative features extracted from strain curves, using three different echocardiographic views.

## Methods

### Population

161 patients undergoing CRT according to current recommendations [20] were included in this multicenter study. Patients were enrolled at the University Hospital of Rennes, at the University Hospital of Oslo, and the Universitair Ziekenhuis Leuven. All patients underwent 2D- standard transthoracic echocardiography before CRT implantation and at 6-month follow-up. Because of the poor-quality of some strain signals and/or irregular heart rhythm, 31 patients were withdrawn from the study, so the final study population was composed of 130 patients. The study was reviewed by an independent ethics committee "Ouest V" ethic committee validation number: 35RC14–9767) and conducted in accordance with the 'Good Clinical Practice' Guidelines in the Declaration of Helsinki. All patients provided written informed consent.

Each patient had the implantation performed during the month following echocardiography. When required, patients received an implantable cardiac defibrillator. Responders were defined as having a > 15% decrease in LV end-systolic volume at the 6-month follow-up, as compared with baseline [21, 22].

### Echocardiographic study

All patients had a complete baseline echocardiography before implantation (GE, Vingmed System 7, Ve9, Ve95, Horten, Norway) equipped with a 3S or M5S 3.5-mHz transducer. Two-dimensional, color Doppler, pulsed-wave, and continuous-wave Doppler data were stored on a dedicated workstation (BT12-EchoPAC PC V202.0.0, GE Healthcare, Horten, Norway) and

analyzed according to current guidelines [8] by a certified senior echocardiographer. LV volumes and ejection fraction were calculated using the biplane modified Simpson method. Systolic ejection time was measured by recording aortic flow with pulsed-wave Doppler imaging from the QRS onset to the aortic valve closure.

In order to obtain LV strain curves, two-dimensional gray-scale images were acquired in the standard apical 4-chamber (4ch), 3-chamber (3ch), and 2-chamber (2ch) views, at a frame rate of at least 60 frames/s. Fig 1 shows a schematic representation of such 3 views, as well as the corresponding position and labeling of the 18 myocardial segments. For each view, offline analysis was performed using the software pack described before. A line was traced along the endocardium's inner border in each of the three apical views on an end-systolic frame, and a region of interest was automatically defined between the endocardial and epicardial borders, with global longitudinal strain (GLS) then automatically calculated from the strain in the three apical views [23].

Close attention was paid to the placement of timing markers (onset of the QRS and aortic valve closure), as previously described by our group [12]. The calculated longitudinal strain signals for each segment were exported from the BT10-EchoPAC software. Each file is composed of longitudinal strain time series corresponding to 6 myocardial segments. These files were processed through a custom-made Python script in order to extract a set of features for further processing.

## Extracted features

Several features were extracted from the strain time series, as depicted in Fig 2. The process of obtaining these features is completely automatic, after the application of a standard cardiac strain study. The first phase consists in obtaining the strain curves, which implies the manual segmentation of the LV with the help of BT10-EchoPAC software. Strain curves are automatically calculated and exported. In our method, we also require a manual verification of the instant of aortic valve closure, which is classically performed in echo analyses. The calculation of the strain features from the obtained strain curves is performed in a completely automatic fashion by a custom Python program developed in our team. This automatic feature extraction aspect increases the reproducibility of the results, allows for a massive application on strain datasets and eases the future translation to the clinics.

In order to minimize the estimation error of these features, each strain curve was upsampled to 500 Hz. As performed in previous works, strain values between −5% and 5% were ignored from all calculations [12]. The onset of the QRS ($T_{QRS}$) is used as reference for the calculation of all features.

The first set of features is obtained from standard amplitude and time-domain analysis of the ECG and the available strain time series. Firstly, the aortic valve closure (AVC) instant ($T_{vw,avc}$) was manually annotated for each view $vw$. Fig 2A shows this feature as a black vertical line on top of the strain signals observed from an example 4ch view.

The peak strain value was automatically identified for each segment of each view. It corresponds to the maximum percentage of contraction and was termed $P_{vw}^{sg}$, for view $vw$ and segment $sg$. The instant at which this peak strain value is identified is represented as $T_{vw,peak}^{sg}$. In Fig 2A, an example of these features for the Basal Septal (BS) segment is presented in red. The mean and standard deviation (Std) of the peak values and their corresponding time instants were calculated for each view and noted $P_{vw}^{Mean}$, $P_{vw}^{Std}$, $T_{vw,peak}^{Mean}$, and $T_{vw,peak}^{Std}$, respectively. The difference between the maximum and the minimum for the two types of features was also calculated

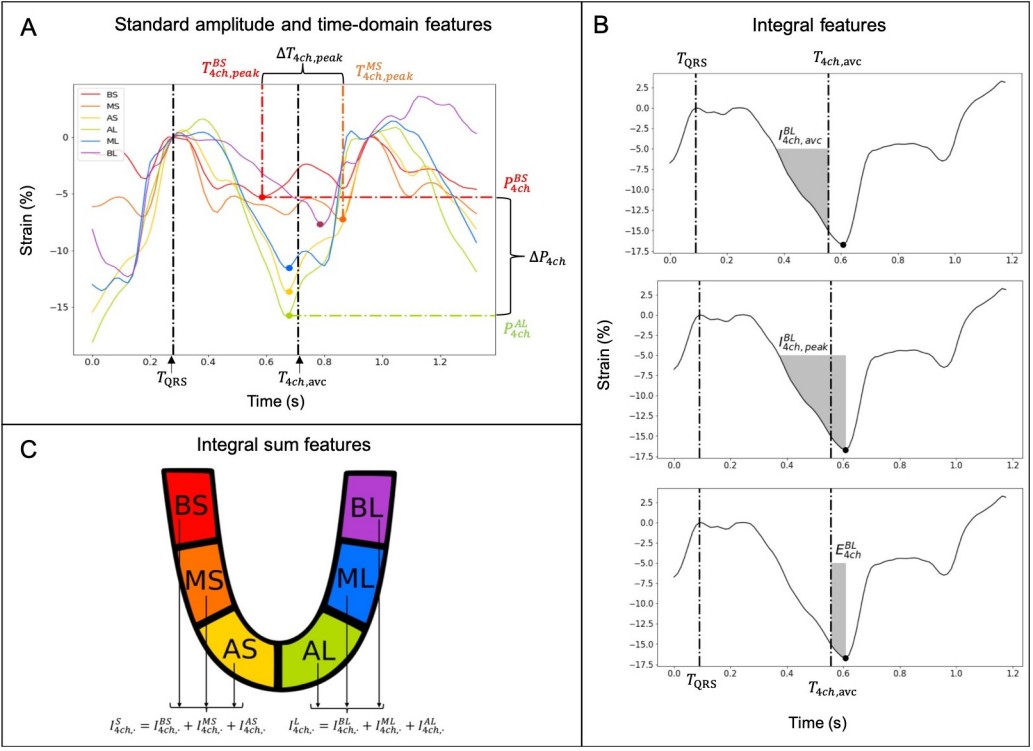

**Fig 2. Feature extraction from longitudinal strain time series in an example case of a 4ch view.** A) Longitudinal strain signals for all segments, observed from a 4ch view, showing standard amplitude and time-domain features extracted from these signals. Black vertical lines correspond to the manually identified onset of the QRS ($T_{QRS}$) and the aortic valve closure instant ($T_{4ch,avc}$). The color dots correspond to the peak strain value ($P_{4ch}^{sg}$) of each segment, arriving at time $T_{4ch,peak}^{sg}$. B) Calculation of three strain integral features from the longitudinal strain signal of segment BL of the 4ch view. C) Identification of the different segments on the 4ch view, showing how the sum of integrals for different segments are calculated (B: basal, M: Mid, and A: Apical, S: Septal and L: Lateral). Note that corresponding colors for each segment were used for panels A and C but these colors are different from those used in Fig 1.

as follows (see Fig 2A):

$$\Delta P_{vw} = \max_{sg}(P_{vw}^{sg}) - \min_{sg}(P_{vw}^{sg}) \tag{1}$$

$$\Delta T_{vw,peak} = \max_{sg}(T_{vw,peak}^{sg}) - \min_{sg}(T_{vw,peak}^{sg}) \tag{2}$$

$\Delta P_{vw}$ is a marker of the heterogeneity of contraction between the segments of one view, while $\Delta T_{vw,peak}$ is a marker of the dyssynchrony between the segments.

The second set of features is based on the estimation of the integral of strain signals, as proposed in our previous works [12]. For each segment, different integrals of the negative part of the strain signal were calculated. Fig 2B shows an example of such extracted integral features for the Basal Lateral segment (BL) of a 4ch view. One integral feature ($I_{vw,avc}^{sg}$) is calculated from $T_{QRS}$ to $T_{vw,avc}$ (upper panel of Fig 2B). It represents the cumulative myocardial strain that is useful to eject blood. A second integral ($I_{vw,peak}^{sg}$) is calculated from $T_{QRS}$ to $T_{vw,peak}^{sg}$ (middle panel of Fig 2B). It represents the total cumulative myocardial strain deployed by the segment. The third integral is calculated as follows:

$$E_{vw}^{sg} = I_{vw,peak}^{sg} - I_{vw,avc}^{sg} \tag{3}$$

and corresponds thus to the integral between the strain peak and the aortic valve closure (lower panel of Fig 2B). This is a marker of the mechanical efficiency of the segment. Positive values of this marker reflect an inefficient (or wasted) cumulative strain, acting after the closure of the aortic valve, when blood is no longer ejected.

This procedure was applied to all segments and to all views. Then, the mean and the standard deviation (Std) of these different integrals were calculated for each view: $I_{vw,avc}^{Mean}$, $I_{vw,avc}^{Std}$, $I_{vw,peak}^{Mean}$, $I_{vw,peak}^{Std}$, $E_{vw}^{Mean}$, and $E_{vw}^{Std}$.

The last set of features was based on sums of integrals [12]. For the two sides $sd$ of each view and for the three different types of integrals, the sums of integrals of the three segments were calculated as follows:

$$I_{vw,avc}^{sd} = \sum_{sg \in sd} I_{vw,avc}^{sg} \tag{4}$$

$$I_{vw,peak}^{sd} = \sum_{sg \in sd} I_{vw,peak}^{sg} \tag{5}$$

$$E_{vw}^{sd} = \sum_{sg \in sd} E_{vw}^{sg} \tag{6}$$

They represent the cumulative strain from the totality of side $sd$. The two possible opposing sides are S = Septal and L = Lateral for 4ch, I = Inferior and A = Anterior for 2ch, As = Anteroseptal and P = Posterior for 3ch. Fig 2C shows, as an example, the two opposing sides in the case of a 4ch view. Finally, the differences of the cumulative strain of the two sides were calculated as follows:

$$I_{vw,avc}^{sd1-sd2} = I_{vw,avc}^{sd1} - I_{vw,avc}^{sd2} \tag{7}$$

$$I_{vw,peak}^{sd1-sd2} = I_{vw,peak}^{sd1} - I_{vw,peak}^{sd2} \tag{8}$$

$$E_{vw}^{sd1-sd2} = E_{vw}^{sd1} - E_{vw}^{sd2} \tag{9}$$

In addition to the above-mentioned features, two additional features that are manually identified by the clinicians have also been studied: the QRS duration and the LVEF. The final set of features extracted from data acquired from a given patient is thus constituted of 158 elements, represented in Table 1.

### Estimation of feature importance

The objective here is to analyze the relative importance of the extracted 158 features for the estimation of the response to CRT. Different methods have been applied and compared in order to estimate this feature importance:

**Random forest-based feature importance estimation.** Feature importance estimation can be performed within the ML chain by comparing the sensitivity of classification performance when applying the selected ML model, using different feature sets [24]. In this paper we used a ML approach based on random forests (RF), a versatile supervised ensemble learner method.

We have chosen to apply RF in this case for two main reasons. Firstly, RF can easily cope with unbalanced classes through a class weighting approach. Secondly, our dataset is of

**Table 1. Number of features per view and in total.**

| Feature | Number per view | Total number | Automatic or manual extraction |
|---|---|---|---|
| $T_{vw,avc}$ | 1 | 3 | Manual |
| $P^v_{vw}$ | 8 | 24 | Automatic |
| $T^v_{vw,peak}$ | 8 | 24 | Automatic |
| $\Delta P_{vw}$ | 1 | 3 | Automatic |
| $\Delta T_{vw,peak}$ | 1 | 3 | Automatic |
| $I^\varsigma_{vw,avc}$ | 11 | 33 | Automatic |
| $I^\varsigma_{vw,peak}$ | 11 | 33 | Automatic |
| $E^\varsigma_{vw}$ | 11 | 33 | Automatic |
| *QRS duration* | | 1 | Manual |
| *LVEF* | | 1 | Manual |

The last column shows if the subset is automatically or manually obtained. Symbols $v \in \{sg, Mean, Std\}$ and $\varsigma \in \{sg, Mean, Std, sd1, sd2, sd1 - sd2\}$.

relatively limited size. RF, with reduced degrees of freedom with respect to other ML methods, makes it an appropriate method for datasets such as the one studied in this work.

In this work, a random forest composed of *M* decision trees has been retained. Trees are configured for classification of each patient in one of two classes: "non-responder" or "responder" and the majority rule between trees is used for the final classification decision. All the available features from all patients were used as input to the RF-based feature importance analysis, which is based on an "out-of-bag" (OOB) approach (Fig 3). In OOB, a bootstrap

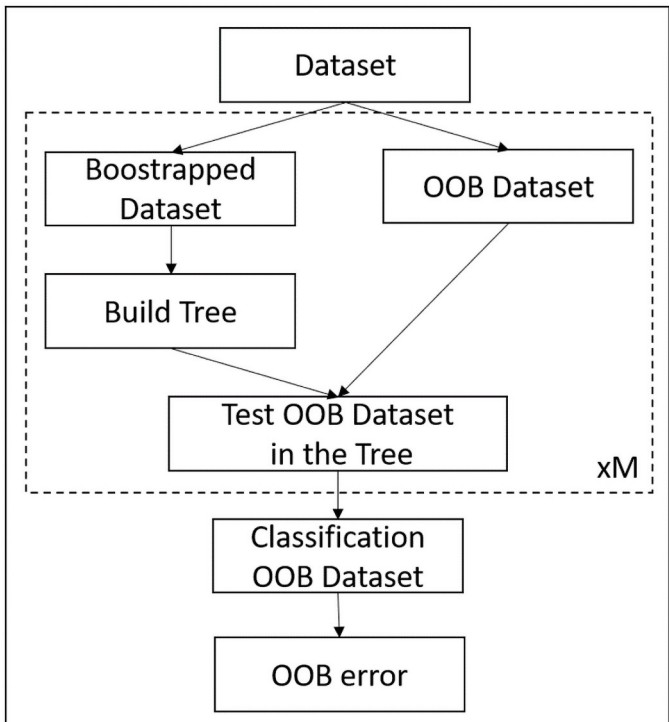

**Fig 3. Out-of-bag process.** The procedure inside the dotted rectangle is performed for each one of the *M* trees on the ensemble.

procedure based on uniform random selection with replacement is applied to create a different learning set for each tree of the RF. Features are also chosen randomly for the construction of each tree. The data that was excluded from the learning set of each tree (out-of-bag data), is used to estimate the performance of the RF. Since the set of features used for the construction of each tree is different, due to the random selection process, the relative sensitivity of each feature to the final decision can also be estimated from this OOB approach.

**Correlation.** The linear correlation between each pair of features has been analyzed by estimating the covariance matrix. The absolute value of the correlation was analyzed for the most important features obtained from the previous analysis step. Only a correlation higher than 0.5 has been taken into account for the analysis.

**Wrapping and filtering methods.** As a complement to the OOB method described above, wrapper and filter methods [18] have also been applied. Filtering approaches use a ranking criterion to remove features below a suitable threshold. For example, the "correlation with the target" filter removes the features for which the correlation with the response to CRT is the lowest. Wrapper approaches evaluate the features' performance to estimate the response to CRT.

We used five wrapping and filtering methods: i) correlation with target, ii) Welch's t-test, iii) K-Best, iv) Recursive Feature Elimination (RFE), and v) Relief. Results from these five methods were combined to produce a single list of relative feature importance. The combination of these results, named CWF in this paper was performed as follows. We firstly created a list of features appearing in the 20 most important features of at least three wrapping or filtering methods. Then, for each feature in this list, the relative importance rank was calculated as the mean importance rank obtained for this feature in the 20 most important feature of the five wrapping and filtering methods. We compared the feature importance of the CWF method and the ML approach.

## Results

### Characteristics of the clinical data

Table 2 shows the baseline characteristics of the population included in the study. The number of women is expressed in number and in percentage according to CRT response. The other characteristics are given in mean and standard deviation. Continuous variables are compared using the Wilcoxon test, categorical data are compared by the $\chi^2$ test. A value of $p < 0.05$ was considered statistically significant.

### Feature importance

After a first stage of local sensitivity analysis, the random forest used for OOB feature importance analysis was configured with $M = 700$ trees, using the 'gini' criteria, with a maximum number of features per tree of 10 and unlimited tree depth [25, 26].

**Table 2. Clinical features.**

|  | Responders 83 (64%) | Non-Responders 47 (36%) | P Value |
|---|---|---|---|
| Women, n (%) | 29 (35%) | 6 (13%) | 0.036 |
| Age (years) | 67.1±10.5 | 65.6±12.3 | 0.576 |
| LVEF (%) | 29.3±7.2 | 28.4±9.7 | 0.434 |
| QRS (ms) | 168.4±19.2 | 165.4±22.0 | 0.293 |

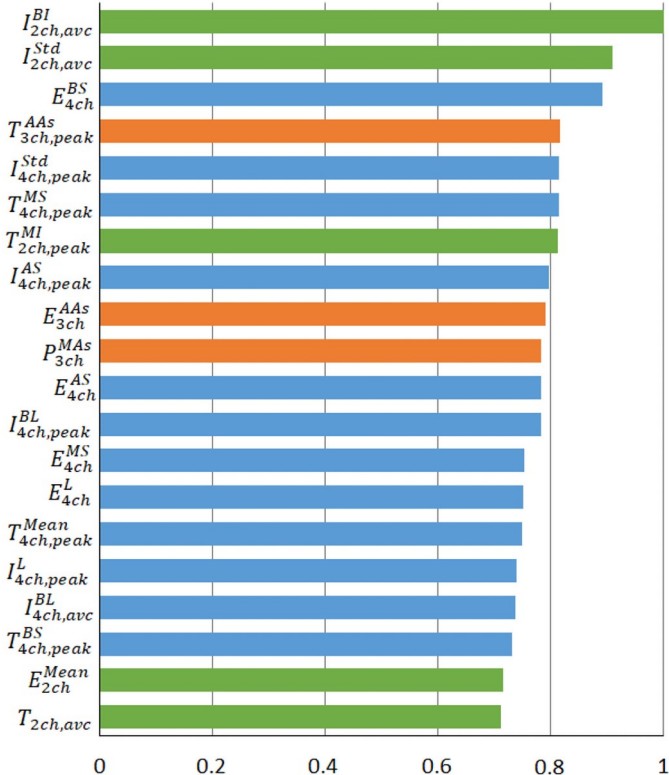

**Fig 4. 20 most important features obtained from the OOB analysis (blue: 4ch, orange: 3ch, and green: 2ch).**

Fig 4 shows the 20 most important features obtained from the OOB analysis. Results were normalized to have the most important feature equal to 1. Most of these significant features (12/20) are obtained from 4ch view, 5/20 from 2ch view, and 3/20 from 3ch view.

All data are available in S1 Data.

## Correlation

Fig 5 indicates the correlation between the 20 most important features. Two given features with an absolute correlation higher than 0.5 are linked by a line. The thickness of the line is calculated as *Thickness* = 10 * *Correlation* − 5. Therefore, lines corresponding to a correlation equal or less than 0.5 will not be shown. A correlation of 1 will produce a line of thickness of 5 points.

## Wrapping/filtering methods and CWF approach

Table 3 shows the 20 most important features obtained from the OOB and CWF methods. In this table, the features are sorted by importance for each method. The features in bold are the common features from both methods. 10 of the 20 most important features of the OOB analysis are also present in the 20 most important ones of the CWF method.

## Discussion

The main objective of this work was to identify the most informative echocardiographic features and views for the estimation of CRT response in terms of LV remodeling. Different feature selection methods were applied and combined in order to obtain a quantitative

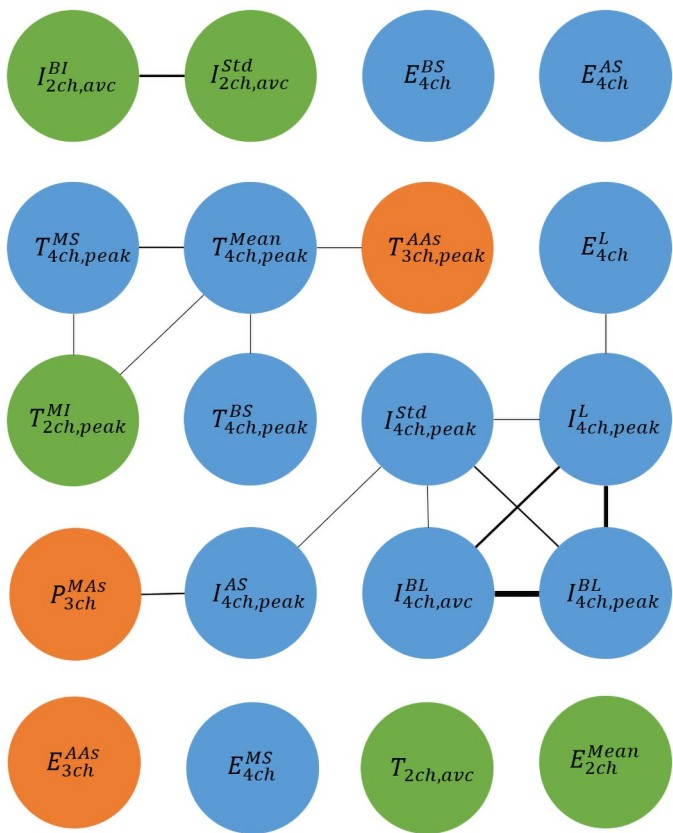

**Fig 5. Correlation between the 20 most important features (blue: 4ch, orange: 3ch, and green: 2ch).** The thickness of the line represents the importance of the correlation between the two features. Only correlation coefficients higher than 0.5 are displayed.

estimation of the relative importance of each feature and view. To our knowledge, this is the first quantitative analysis of the contribution of multi-view echocardiographic features to the estimation of CRT response. This could be a step forward for CRT that remains under-used, partly because of the problem of lack of response of certain patients and the imperfect selection criteria that are recommended, up to now, by current guidelines [3].

**Table 3. List of the 20 most important features estimated using OOB and CWF.**

| Rank | OOB | CWF |
|---|---|---|
| 1 to 3 | $I^{BI}_{2ch,avc}$, $I^{Std}_{2ch,avc}$, $E^{BS}_{4ch}$, | $I^{BI}_{2ch,avc}$, $P^{AAs}_{3ch}$, $E^{AA}_{2ch}$, |
| 4 to 6 | $T^{AAs}_{3ch,peak}$, $I^{Std}_{4ch,peak}$, $T^{MS}_{4ch,peak}$, | $T^{Mean}_{4ch,peak}$, $E^{MS}_{4ch}$, $E^{BS}_{4ch}$, |
| 7 to 9 | $T^{MI}_{2ch,peak}$, $I^{AS}_{4ch,peak}$, $E^{AAs}_{3ch}$, | $E^{Mean}_{2ch}$, $P^{AS}_{4ch}$, $T^{MI}_{2ch,peak}$, |
| 10 to 12 | $P^{MAs}_{3ch}$, $E^{AS}_{4ch}$, $I^{BL}_{4ch,peak}$, | $I^{AA}_{2ch,peak}$, $I^{L}_{4ch,avc}$, $I^{AS}_{4ch,avc}$, |
| 13 to 15 | $E^{MS}_{4ch}$, $E^{L}_{4ch}$, $T^{Mean}_{4ch,peak}$, | $I^{L}_{4ch,peak}$, $E^{BI}_{2ch}$, $T^{AAs}_{3ch,peak}$, |
| 16 to 18 | $I^{L}_{4ch,peak}$, $I^{BL}_{4ch,avc}$, $T^{BS}_{4ch,peak}$, | $E^{I}_{2ch}$, $E^{MA}_{2ch}$, $P^{MAs}_{3ch}$, |
| 19 and 20 | $E^{Mean}_{2ch}$, $T_{2ch,avc}$ | $T^{MS}_{4ch,peak}$, $E^{MI}_{2ch}$ |

Features are sorted by importance. Common features from both methods are marked in bold.

This paper was focused on a data processing pipeline, based on the explicit extraction of a set of physiologically interpretable features, and the application of Random Forest as the underlying model. Another possible approach would have been to apply Deep Learning (DL) to the raw image data. We have clearly avoided such a DL approach because of several reasons [27, 28]. DL lacks interpretability and this aspect seems to us incompatible with the targeted clinical application. Moreover, DL, characterized by a particularly high number of degrees of freedom, is highly sensitive to noise, to overfitting and requires massive datasets. Our choice for a features-based approach was considered more adapted to this problem.

In the OOB analysis, the 4ch view provides the highest proportion of important features, with 60% of them in the 20 most important features. When we consider the 20 most important features, half of the uncorrelated features (3/6) are obtained from the 4ch view and we observe a low correlation between features within the 4ch view (between 0.5 and 0.6). Results from the CWF method also show many features from the 4ch view (40%). We can also notice that 50% of the significant features that have been selected through both feature selection methods are observed from the 4ch view. These results suggest that the 4ch view provides the highest proportion of information for the prediction of CRT response and should be considered as the main view to prioritize if only one view can be analyzed. Studies have already demonstrated that the value of 4ch view is of primary importance in this context [29].

Nevertheless, from the OOB analysis, 40% of the 20 most important features are obtained from the two other views. Features from the 2ch view represent 25% of the most important features and, moreover, the two most important features overall are obtained from the 2ch view ($I_{2ch,avc}^{BI}$ and $I_{2ch,avc}^{Std}$). However, these two features are correlated (0.63). The correlation analysis shows that only one feature of the 2ch view, $T_{2ch,peak}^{MI}$, is correlated with two other time domain features from the 4ch view. Features from the 2ch view are also very important according to the CWF method (45%). These facts show that the 2ch view can provide important complementary information for the prediction of the CRT response and should be the second view to prioritize. The 3ch view shows the lowest proportion of important features (15%) and the selected features from this view were usually correlated with features from the 4ch view. The 3ch view can thus provide rather low additional information and should be considered as the lesser priority.

The 20 most important features from the OOB and CWF analyses include features of different subsets, mainly time-domain and strain integrals. Correlation often appears for features from the same subset and rarely between different subsets. The regional heterogeneity of the features extracted from strain integrals appears as a particularly useful marker for the prediction of CRT response. Almost every cardiac segment, except those from the 3ch, appear in at least one of the selected features.

In our analysis combining OOB and CWR results, we identified 10 main features that are particularly informative to predict CRT response. 8 of these features are extracted from the 4ch view, and 2 from the 2ch view. $I_{2ch,avc}^{BI}$, and $I_{2ch,avc}^{Std}$ obtained from the 2ch view are two most important features. $I_{2ch,avc}^{BI}$ represents a quantification of the cumulative strain developed by the infero-basal segment, which effectively contributes to LV ejection, whereas $I_{2ch,avc}^{Std}$ estimates the heterogeneity of the cumulative strain developed by the anterior and inferior LV wall during contraction. To our knowledge, the specific role of 2ch-derived features in the prediction of CRT response has never been described before. Nevertheless, the cumulative strain observed during LV contraction, which is an expression of myocardial viability, is known to be directly correlated to CRT response and survival after CRT [30, 31] and might contribute to explain the effect of $I_{2ch,avc}^{BI}$ on CRT efficacy. Also, no previous publication has addressed the role of opposite wall activation in 2ch view for the prediction of CRT response. Yu et al. have shown

that the standard deviation of opposite wall delay measured in short axis view, which includes the assessment of the anterior and inferior segments, is an interesting marker to predict CRT response [32]. The mechanical dyssynchrony of the anterior and posterior LV and the subsequent heterogeneity of strain signals can explain the importance of $I_{2ch,avc}^{Std}$ in CRT.

$E_{4ch}^{BS}$, $E_{4ch}^{AS}$ and $E_{4ch}^{MS}$ are respectively the third, the eleventh and the thirteenth most important features. They are the expression of the mechanical efficiency of the septal segments. In normal subjects, all LV segments contract almost simultaneously. In patients with left bundle branch block, which are typical candidates to CRT, the ventricular conduction delay causes the early activation of the septum and a delayed activation of the lateral wall. In these patients, the early septal activation is energetically inefficient because the LV pressure is still low at this early instant of the cardiac cycle and the aortic valve is still closed. Thus, this mechanical activation does not contribute to LV ejection [33]. Previous studies have shown that the septal wasted energy is increased in CRT candidates and is useful to predict CRT response [34, 35].

$I_{4ch,peak}^{AS}$ is the eighth most important feature in the OOB analysis. This feature represents the total strain deployed by the AS segment. According to Cikes et al. [13], strain is low in the apical region for non-responders to CRT. This makes sense, since AS is often one of the septal segments where the rebound stretch is the highest. In other words, this is one of the segments where the lack of efficiency of myocardial systolic strain is the greatest.

$I_{4ch,peak}^{L}$ is the sixteenth most important feature in the OOB classification and thirteenth in the CWF. In patients undergoing CRT, the delayed activation of the lateral wall lasts after aortic valve closure. This means that only a part of the cumulative strain developed by the lateral wall will contribute to LV ejection (Fig 2B). $I_{4ch,peak}^{L}$ corresponds to the global cumulative strain developed by the contraction of the lateral wall and is an indirect marker of lateral wall viability. Because the electrical stimulation of the lateral wall is a target of CRT therapy, the amount of mechanical energy which can be developed by this wall is known to be associated to CRT response [36, 37].

Other well-ranked 4ch-derived variables include $T_{4ch,peak}^{Mean}$, $T_{4ch,peak}^{MS}$ and $T_{4ch,peak}^{BS}$. $T_{4ch,peak}^{MS}$ and $T_{4ch,peak}^{BS}$ represent the time necessary to the maximal mechanical activation of the corresponding septal segments. The true LBBB mechanics causes a significant activation delay between the septal and the lateral wall. The presence of this typical pattern, with early septal activation and delayed lateral wall contraction [38] can be reversed by CRT and can be associated with significant reverse LV remodeling. It might explain the predictive role of septal activation timing. $T_{4ch,peak}^{Mean}$ is a measure of the global LV mechanical discoordination, which can be assessed in 4-chamber view. Previous studies have demonstrated that the heterogeneity of activation timings of opposite LV walls [39, 40], or the standard deviation of opposite walls delay are interesting predictors of CRT response. Nevertheless, these parameters are plagued by a low reproducibility when calculated manually and their utility has never been demonstrated in large multicenter trials, which can explain the lower predictive value of these variables [41].

From this discussion, it appears that a multivariate analysis of regional myocardial strain information is an added value for the prediction of the response to CRT. Cikes et al. have shown some similar results [13] through unsupervised machine learning. Although only qualitative strain analyses were performed in their work, they have pointed out the importance of the strain shape on segments AS, BS, and BI for the prediction of CRT outcome. In our study, quantitative strain analysis yields results supporting this fact, since features $I_{4ch,peak}^{AS}$, $E_{4ch}^{AS}$, $E_{4ch}^{BS}$, and $I_{2ch,avc}^{BI}$ are considered informative. We find the same results for the AA segment in the CWF method ($E_{2ch}^{AA}$) but not in the OOB method. Nevertheless, we did not find the same results for the AL segment, except in the sum of integrals ($I_{4ch,peak}^{L}$). More importantly, as described

above, we have identified a number of other particularly informative features that should be considered for the estimation of CRT response and the proposed method is based on a quantitative approach that may be easily included into a machine learning pipeline.

## Limitations

Despite its multicenter design, this study included a limited number of patients. Some of them had to be withdrawn because of atrial fibrillation or poor quality strain signals. The quality of images and the presence of sinus rhythm have a pivotal importance for the assessment of strain dynamics, but cannot be obtained for all patients in clinical practice.

Moreover, this study focuses on patients that are supposed to be implanted according to guidelines (I-A). Patients with irregular heart rhythm were excluded of this study. In a future work, these results could be extended to those patients.

A higher number of patients is necessary to study the performance of the RF to predict CRT response. The performance could be studied for the best features per view and for all views.

## Conclusion

In the present study, we applied different feature analysis methods to characterize a pool of quantitative features extracted from different views and methods from cardiac echocardiographic analyses. We found that most of the significant strain-derived features for the prediction of CRT response are obtained from the 4ch view. According to our data, the assessment of strain dynamics in 4ch view provide the most important information to predict CRT response. The 2ch view, might provide additional, complementary information on LV deformation. This analysis is a first step towards an improved development of multivariate machine learning methods for CRT prediction.

## Supporting information

**S1 Data. Data disclosure.** Each line corresponds to a different patient and each column to a feature.
(XLSX)

## Acknowledgments

The authors would like to thank Valerie Le Moal and Isabau Labrousse for their skillful assistance in the collection of data.

## Author Contributions

**Conceptualization:** Alban Gallard, Erwan Donal, Alfredo I. Hernández.

**Data curation:** Alban Gallard, Elena Galli, Arnaud Hubert, Auriane Bidaut, Virginie Le Rolle, Otto Smiseth, Jens-Uwe Voigt, Erwan Donal, Alfredo I. Hernández.

**Formal analysis:** Alban Gallard, Alfredo I. Hernández.

**Investigation:** Elena Galli, Arnaud Hubert, Auriane Bidaut, Otto Smiseth, Jens-Uwe Voigt, Erwan Donal.

**Methodology:** Alban Gallard, Virginie Le Rolle, Alfredo I. Hernández.

**Project administration:** Erwan Donal, Alfredo I. Hernández.

**Resources:** Elena Galli, Arnaud Hubert, Auriane Bidaut, Otto Smiseth, Jens-Uwe Voigt, Erwan Donal.

**Software:** Alban Gallard, Virginie Le Rolle, Alfredo I. Hernández.

**Supervision:** Erwan Donal, Alfredo I. Hernández.

**Validation:** Alban Gallard, Virginie Le Rolle, Erwan Donal, Alfredo I. Hernández.

**Visualization:** Alban Gallard.

**Writing – original draft:** Alban Gallard.

**Writing – review & editing:** Alban Gallard, Elena Galli, Arnaud Hubert, Erwan Donal, Alfredo I. Hernández.

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
