## [Decision Letter · Decision Letter 0]

5 May 2021

PONE-D-21-10579

Echocardiographic view and feature selection for the estimation of the response to CRT

PLOS ONE

Dear Dr. Alban Gallard,

Thank you for submitting your manuscript to PLOS ONE. After careful consideration, we feel that it has merit but does not fully meet PLOS ONE’s publication criteria as it currently stands. Therefore, we invite you to submit a revised version of the manuscript that addresses the points raised during the review process.

Please submit your revised manuscript by 19/05/2021, looking for reviewers' comments and criticisms.

If you will need more time than this to complete your revisions, please reply to this message or contact the journal office at plosone@plos.org. Please include the following items when submitting your revised manuscript:

We look forward to receiving your revised manuscript.

Kind regards,

Giuseppe Coppola

Academic Editor

PLOS ONE

Journal Requirements:

3. Thank you for including your ethics statement:

“ The study was reviewed by an independent ethics committee (regional ethic committee validation number: 35RC14-9767) and conducted in accordance with the ‘Good Clinical Practice’ Guidelines in the Declaration of Helsinki. All patients provided written informed consent. “

**Comments to the Author**

1. Is the manuscript technically sound, and do the data support the conclusions?

Reviewer #1: Yes

Reviewer #2: Yes

2. Has the statistical analysis been performed appropriately and rigorously? 

Reviewer #1: Yes

Reviewer #2: Yes

3. Have the authors made all data underlying the findings in their manuscript fully available?

Reviewer #1: Yes

Reviewer #2: Yes

4. Is the manuscript presented in an intelligible fashion and written in standard English?

Reviewer #1: Yes

Reviewer #2: Yes

5. Review Comments to the Author

Reviewer #1: The article is original, well written. It analyzes the use of strain in the prediction of CRT responders.

Over time, different methods and different echocardiographic criteria have been used in predicting the response to CRT. In the introduction the authors may cite the review "Non-responders to cardiac resynchronization therapy: Insights from multimodality imaging and electrocardiography. A brief review" published in International Journal of Cardiology (doi: 10.1016 / j.ijcard.2016.09.037) and other articles related to old and new echocardiographic methods useful in identifying patient responders to CRT.

Reviewer #2: The authors investigated the relative importance of strain-based features obtained from the 4ch, 3ch and 2ch apical views in identifying patients responders to cardiac resynchronization therapy. They found 20 features as the most important, 60% of them obtained from the 4ch apical view. The study is interesting, well written and conducted. However I have some comments:

1) How do you translate your findings in routine clinical practice? You identified 20 features as the the most important, however some of them, particularly those derived from integrals, seem not easy to obtain and/or time-consuming, thus their application in clinical practice seem doubtful. Please comment it.

2) Figure 2 should be improved. The text in the figure is not legible.

3) Table 3 is a little bit confusing. Please improve it.

6. PLOS authors have the option to publish the peer review history of their article (what does this mean?). If published, this will include your full peer review and any attached files.

Reviewer #1: No

Reviewer #2: No

---

## [Author Response · Author response to Decision Letter 0]

21 May 2021

We attached a file named "Response to Reviewers" to respond to the different comments.

---

## [Editor Report · Decision Letter 1]

25 May 2021

Echocardiographic view and feature selection for the estimation of the response to CRT

PONE-D-21-10579R1

Dear Dr. Alban Gallard,

We’re pleased to inform you that your manuscript has been judged scientifically suitable for publication and will be formally accepted for publication once it meets all outstanding technical requirements.

Kind regards,

Giuseppe Coppola

Academic Editor

PLOS ONE

---

## [Editor Report · Acceptance letter]

1 Jun 2021

PONE-D-21-10579R1 

Echocardiographic view and feature selection for the estimation of the response to CRT 

Dear Dr. Gallard:

I'm pleased to inform you that your manuscript has been deemed suitable for publication in PLOS ONE. Congratulations! Your manuscript is now with our production department. 

Kind regards, 

on behalf of

Dr. Giuseppe Coppola 

Academic Editor

PLOS ONE